# Efficacy of Sea Salt-Based Mouthwash and Xylitol in Improving Oral Hygiene among Adolescent Population: A Pilot Study

**DOI:** 10.3390/ijerph18010044

**Published:** 2020-12-23

**Authors:** Andrea Ballini, Stefania Cantore, Luca Signorini, Rajiv Saini, Salvatore Scacco, Antonio Gnoni, Alessio Danilo Inchingolo, Danila De Vito, Luigi Santacroce, Francesco Inchingolo, Gianna Dipalma

**Affiliations:** 1Department of Biosciences, Biotechnologies and Biopharmaceutics, Campus Universitario “Ernesto Quagliariello”, University of Bari Aldo Moro, 70125 Bari, Italy; andrea.ballini@uniba.it; 2Department of Precision Medicine, University of Campania “Luigi Vanvitelli”, 80100 Naples, Italy; 3Department of Interdisciplinary Medicine, University of Bari Aldo Moro, 70124 Bari, Italy; editorijeds@gmail.com (R.S.); luigi.santacroce@uniba.it (L.S.); francesco.inchingolo@uniba.it (F.I.); giannadipalma@tiscali.it (G.D.); 4Dental School, Unicamillus University, 00131 Rome, Italy; dottluca.signorini@libero.it; 5Department of Basic Medical Sciences, Neurosciences and Sense Organs, University of Bari Aldo Moro, 70124 Bari, Italy; antonio.gnoni@uniba.it (A.G.); danila.devito@uniba.it (D.D.V.); 6School of Technical Medical Sciences, “A. Xhuvani” University, 3001 Elbasan, Albania

**Keywords:** xylitol, sea salt, lysozyme, plaque index, oral health, *Streptococcus mutans*, clinical microbiology

## Abstract

The scientific community has definitely demonstrated the importance of the use of mouthwash in daily oral hygiene. In our pilot study, we tested the effectiveness of a novel mouth rinse containing sea salt, xylitol, and lysozyme. *Streptococcus mutans (S. mutans)* growth, and plaque index in adolescent patients aged 14–17 years, were observed. The bacterial load was investigated by in vitro microbiological analysis; the plaque index was assessed through the O’Leary’s Plaque Control Record (PCR). The study has shown that the use of a sea salt-based mouthwash in daily oral hygiene reduces the bacterial levels of *S. mutans* (*p* < 0.01) linked to the combined action of xylitol and lysozyme, together with the action of sea salt. Our preliminary data confirm and improve the main results reported in the scientific literature on the importance of the use of xylitol, lysozyme, and sea salt in oral health.

## 1. Introduction

Caries is the oral pathology with the highest prevalence in developmental age. It is considered a multifactorial disease, and the presence of specific pathogenic bacteria on teeth plays a pivotal role in its etiopathogenesis [1,2,3]. Currently, caries has been demonstrated to be a polymicrobial disease associated with a complex community of microorganisms, made up of hundreds of bacterial species: in this context, *Streptococcus mutans (S. mutans)* and *Lactobacillus acidophilus (L. acidophilus)* have a major role as causative agents of dental caries [1].

The cariogenic potential of *S. mutans* is closely related to its ability to adhere to the tooth surface and to form acids by fermenting sugars introduced with food [4]. The production of acids, in particular lactic acid, is a very important virulence factor of *S. mutans* which leads to the multiplication of bacteria [1,4]. The sugars are fermented by the bacteria of the dental plaque thus producing a particularly acidic environment [4]. The lowering of the pH determined by the germs of the bacterial plaque is sufficient to decalcify the enamel and/or dentine thus creating the first step towards the development of the carious process [4]. Furthermore, the excessive and continuous intake of sugar with the diet allows *S. mutans* to synthesize intracellularly glycogen-like polysaccharides which are used by the bacteria themselves to produce acids when the external sugar intake is limited [1]. *Lactobacilli* have also significantly increased in carious lesions, but they should be considered as “secondary invaders” since it has been seen that, by themselves alone, they are not capable of producing caries [5]. *Lactobacilli* find favorable conditions for their development in the low pH environment created by *S. mutans* and, by developing, contribute to the progression of the carious process [5].

Other studies confirm how proper nutrition, including probiotics, and proper feeding practice from infants can help prevent the onset of carious lesions which will then form in permanent teeth [6,7,8,9,10]. Early disease prevention as well as proper oral hygiene at any age is essential. In fact, both caries and periodontitis are often linked to insufficient levels of oral hygiene, as both have bacterial plaque as the common etiological factor [11,12].

Dental plaque-induced oral dysbiosis and associated diseases, including gingivitis and periodontal disease, has a global distribution and includes a range of diseases that can be mild and/or potentially lethal with the passage of time, also in conjunction with other pathological factors [13,14,15,16,17,18,19]. Among the bacteria that populate the oral cavity, some are associated precisely with the onset of periodontitis, these are called periopathogens. The pathogens involved in periodontal disease are *Aggregatibacter actinomycetemcomitans, Porphyromonas gingivalis, Prevotella intermediate, Tannerella forsythia, Treponema denticola, Prevotella nigrescens, Capnocytophaga sputigena, Eikenella corrodens, Fusobacterium nucleatum, Campylobacter rectus* [19,20]. The periodontal disease manifests itself differently in adolescents and adults. The progression and severity with which the disease presents itself is also different [12,21,22]. The main risk factor for caries and periodontitis in adolescents, according to recent studies, is the calcified presence of bacterial biofilm, a phenomenon that leads to inflammation of the gums as reported from translational research [13,14,15,16,19]. Chlorhexidine and xylitol have been used as strategies to prevent and reduce plaque accumulation [6,11] and new alternative practices are advancing, such as the simultaneous use of sea salt and xylitol [23].

The aim of this study was to evaluate the efficiency of a combined mouth rinse containing purified water, sea salt, xylitol, lysozyme, and menthol (*H2Ocean Sea Salt Mouthwash*, USA) in the management of biofilm formation and gingival health in a group of young adults.

## 2. Study Design

### 2.1. Participants

This research was conducted in collaboration with the Elbasan University “A. Xhuvani” (School of Technical Medical Sciences), Elbasan, Albania, a dental community cabinet (Sorriso & Benessere—Ricerca e Clinica SRL, Bari, Italy), and the University of Bari Aldo Moro, Bari, Italy, as a randomized, double-blinded placebo-controlled study. The Institutional Ethics Committee of the Faculty of Technical Medical Sciences of Elbasan “Aleksandër Xhuvani” has approved the application to conduct the clinical trial in the Faculty. Protocol Identification: INTL_ALITCOOP/DentPath/2020_SLK. The study was carried out according to the Helsinki declaration and informed written consent was obtained from all patients’ parents or guardians included in the study. Twenty healthy adolescent patients aged 14–17 years, were enrolled in the present study. For this pilot, research was designed and planned the experimental protocol according to current literature and based on local epidemiological studies [3,23,24,25,26].

### 2.2. Inclusion and Exclusion Criteria

Potentially eligible participants were identified from the patient record using automated searches, in terms of age. Subsequently, they were screened for inclusion and exclusion criteria and enrolled in the study [23,24,25,26]. Inclusion criteria were age ≤18 years age (14–17 years), good general health, and not under any drug therapies. The exclusion criteria were: a (i) presence of orthodontic treatment (ii) presence of diseases that alter the gingival and salivary health status; (iii) presence of craniofacial or dentofacial anomalies; (iv) uncooperative patient, not available to attend periodic checks; (v) therapy with antibiotics within 2 months before the enrollment; (vi) failure to obtain the informed consent.

### 2.3. Experimental Design

Participants who met the inclusion and exclusion criteria were randomized into one of two groups: supplementation with combined mouthwash (2 rinses daily) or 2 rinses with a placebo (mint-flavored and colored water) for 3 months, as per standard protocol [23,26]. Due to the trademark regulations, the concentrations of single components are not available so we have evaluated the whole product effect on oral bacteria. Written informed consent was obtained from all patients’ parents or guardians included in the study. All patients involved underwent a professional oral hygiene session and instructions for home oral hygiene procedures were given in the conduct described.

On day 1, after disclosing the teeth, the volunteers received an oral soft and hard tissue examination and professional scaling and polishing to remove all calculus, plaque, and extrinsic tooth stain. This will be performed using hand instruments, mechanical scalers, rotating brushes with polishing paste, and dental floss in the interproximal areas. To ensure that all deposits will be removed, if necessary, a second disclosing episode will be carried out. All measurements were carried out under similar clinical settings and by the same calibrated dentists investigators (A.B. and S.C.). All bottles with mouthwash provided to subjects were pre-weighed. All participants were instructed to refrain from using any other means of oral hygiene during the experimental period. All subjects were instructed to rinse twice a day, after tooth brushing, in the morning and at night, for 60 s with 20 mL of solution. Subsequent rinsing with water, drinking, or eating was not allowed for the next hour and prior to saliva collection. The plaque score was recorded prior to saliva collection. All returned mouthwash bottles were weighed to calculate the amount of mouth rinse used and to check for compliance.

### 2.4. Streptococcus mutans Counts (SM) and Plaque Index Evaluation

Patients were evaluated for on *S. mutans* counts (SM) and plaque index after 15 days (T0), and at 1 (T1) and 3 months (T2) after the first rinse [23].

### 2.5. Methods

#### 2.5.1. Microbiological Examination

A sample of at least 150 μL of whole unstimulated saliva (initial volume 2–3 mL) was collected from the 20 participants for microbiological examination [11,23] in the next 60 min subsequent to daily oral hygiene procedures (tooth brushing and oral rinse with the combined or placebo formula), and between the 8:00 and 9:00 a.m., in the Saturday. Parameters for calculating *S. mutans* levels were reported in Table 1. Bacterial colonization was assessed by serial saliva dilution culture method to evaluate the total aerobic microbial load and the possible presence and load of *Streptococcus mutans*. The samples were collected in a sterile tube and diluted in physiological solution to obtain 40-, 800- and 8000-folds the starting concentration. Aliquots of 50 μL of each dilution and the original sample were sown on 2 different culture media: Agar-based blood medium for the evaluation of the microbial load viable and on selective medium *Mitis salivarius* Agar for a culture of *S. mutans*. All plates were incubated in an aerobic atmosphere with 5% CO_2_ at 37 °C for 48 h to visually evaluate the presence of colony-forming units (CFU).

#### 2.5.2. Plaque Index

The plaque index assesses the accurate amount of bacterial plaque in the oral cavity and therefore the general conditions of oral hygiene. The index used was O’Leary’s Plaque Control Record (PCR) which assesses the presence or absence of plaque deposits relative to the gingival margin around the four surfaces of each tooth (facial, lingual, mesial, distal) [27]. The plaque on the four dental surfaces was highlighted with a plaque detector (*GUM^®^ Red-Cote Plaque Disclosing Tablets,* Saronno, Italy), introduced into the oral cavity after brushing, and smeared on the surface of the teeth with the tongue. The plaque index per subject was then calculated as the average of the index of the individual elements. The average is definite as the sum of the indices of the individual dental surfaces with plaque, divided by the number of dental surfaces examined considered by percent.

## 3. Statistical Analysis

Outcome measures of the exploratory study were analyzed with a paired samples non-parametric Wilcoxon test for pre-post differences with time as the factor using R 4.0.3 software, to detect significant differences between pre-test and post-test scores. A *p*-value ≤ 0.05 was set as statistically significant.

## 4. Results

On the first visit, the participants were instructed on the objectives and methods of clinical trials. Subjects who agreed to participate in the study, after having signed the informed consent, were randomly included and divided into two groups: a test group using a combined mouth rinse containing purified water, sea salt, xylitol, lysozyme, and menthol (*H2Ocean Sea Salt Mouthwash*, USA), the other group a placebo oral rinse (mint-flavored and colored water).

Twenty healthy adolescents were enrolled, according to our experience in local epidemiological studies [3], as follows:Placebo Group: 5 males and 5 females.Test group: 6 males and 4 females.

The data obtained for plaque index in the placebo vs. test group at the three different times points, the reference parameters for calculating *S. mutans* levels and related scores, are shown in Table 2 and Table 3.

Figure 1 shows the trend of the plaque index of the test group from T0 to T2. Unlike the placebo group, there is a greater reduction of 21%. The data shows that oral hygiene performed with xylitol and sea salt for 3 months produces positive effects on the plaque.

In addition, Figure 2 shows the trend of the placebo group plaque index from T0 to T2. A reduction of 3% is noted during the mouthwash testing period.

Moreover, Figure 3 shows subjects are in negligible to low caries risk (Class 0–1) with a greater reduction in *S. mutans* levels at 3 months in the test group. This reduction is due to the beneficial effects of xylitol which inhibits bacterial growth. Sea salt, on the other hand, increases the pH balance in favor of an alkaline environment in which bacteria do not survive. Additionally, xylitol has an important cario-preventive activity even in the long term, an activity that is expressed by reducing the concentration of *Streptococci* and consequently through a reduction in the levels of lactic acid produced from T0 to T2.

Figure 4 shows how over the 3 months, *S. mutans* levels changed in the placebo group. Unlike the test group, the subjects performed rinses with a placebo substance which put subjects in moderate to high caries risk (Class 2–3) with moderate reduction, therefore a persistence of the bacterial biofilm.

Finally, the *p*-values of the paired Wilcoxon test are stated in Table 4. As reported, both Test Group Plaque Index (%) and Test Group *S. Mutans* (score) demonstrated a statistically significant difference with *p* < 0.001. Furthermore, no side effects occurred during this pilot study.

## 5. Discussion

In this study, the synergistic effects of a combined mouthwash on oral bacteria accounting for oral pathologies, and more generally on plaque, the bacterial biofilm that forms on dental surfaces, were evaluated. The subjects who followed the experimental protocol saw the plaque indexes and the presence of harmful bacteria in the mouth significantly decrease in favor of greater cleaning and oral hygiene. Checking the plaque on the dental surface is of fundamental importance to prevent the onset of diseases related to poor oral hygiene [17]. This control can be done mechanically with antimicrobial chemical agents, but surely prolonged use would cause undesirable effects [18]. However, an excellent solution is represented by xylitol. According to numerous studies in the literature, xylitol has powerful benefits for dental health and the prevention of tooth plaque [28,29,30,31,32]. Xylitol, a low glycemic sugar, is known to be harmful to many microorganisms and its effect on pneumococci was determined in a 2004 study, showing to decrease the growth of *Streptococcus pneumoniae* (*S. pneumoniae*) in children by 30–42% [30].

More generally, xylitol reduces the levels of *S. mutans* in plaque and saliva by blocking the metabolic processes that induce cellular energy [29]. An experimental study was conducted to understand the useful dosage of xylitol to eliminate *S.mutans* in saliva and plaque. In the experiment were used chewing gums containing xylitol. The subjects participating in the study saw plaque decrease from the dental surfaces: in particular, the results showed a significant reduction of *S. mutans* of plaque after taking 6.44 g of xylitol per day for five weeks [31].

The effect of xylitol can also be enhanced with the synergistic use of other antimicrobial agents. Xylitol combined with chlorhexidine can contribute to the improvement of new formulations of caries preventive mouthwashes [32]. Significant reductions in *S. mutans* scores occurred after using the mouthwash for four weeks. Therefore, researchers have once again confirmed the bacteriostatic effect of xylitol mouthwash on *S. mutans* [32]. In our study, the effect of xylitol was enhanced by the use of sea salt.

According to the data reported by the dental literature, oral saltwater rinses alkalize the oral environment, and an acidic environment helps bacterial growth [32,33,34,35]. Furthermore, rinses with antiseptic mouthwashes have the ability to reduce the infectious quantity in the oral plaque but not to completely eliminate the presence of viruses in the saliva [35]. Our previous findings in the field investigated the ability of sea salt to interact with wound healing processes following oral surgery [26]. From the study carried out, through the randomization of 30 patients, excellent results were emitted. Sea salt accelerated the healing of the surgical site in favor of a highly alkaline environment. The use of sea salt has not produced undesirable effects [26]. Despite some studies ascertaining its validity, however, very few published clinical data remain which attest to its effectiveness within the oral cavity. Micheal et al. noticed a reduction in bacterial biofilm and more generally in plaque indices in those children who used water and salt [33]; investigating in adults with gingivitis, Mani et al. also clarified the efficiency of the oral health of those who used a sea salt mouthwash [34]. Hoover et al. in 2017 published a study according to which, salt brings benefits in subjects with periodontitis but these findings were not significantly relevant [35]. The use for 30 days with water and salt-based rinses alone, without other components, did not affect plaque indices [35]. However, for a more simple and practical daily care, the alleged beneficial effects of combined mouth rinse seem to be a very useful tool to promote oral health. Limitations of this pilot study included the comparatively small sample size and the relatively short trial period.

Based on the information so far in the literature and the traditional use of combined saltwater, xylitol and lysozyme rinses, further clinical studies involving a larger sample size, subjects from a diverse population, and a longer trial period, need to be conducted to investigating its role as a therapeutic agent in the management of daily oral hygiene.

## 6. Conclusions

Based on the results obtained, it appears that rinsing with a combined mouth rinse containing purified water, sea salt, xylitol, lysozyme, menthol, such as *H2Ocean Sea Salt Mouthwash*, USA has a significant advantage on oral health, showing a decrease in plaque index and bacterial charge, lead in benefits until the end of the study period which permits its utilization as an adjunct to daily oral prophylaxis.

## Figures and Tables

**Figure 1 ijerph-18-00044-f001:**
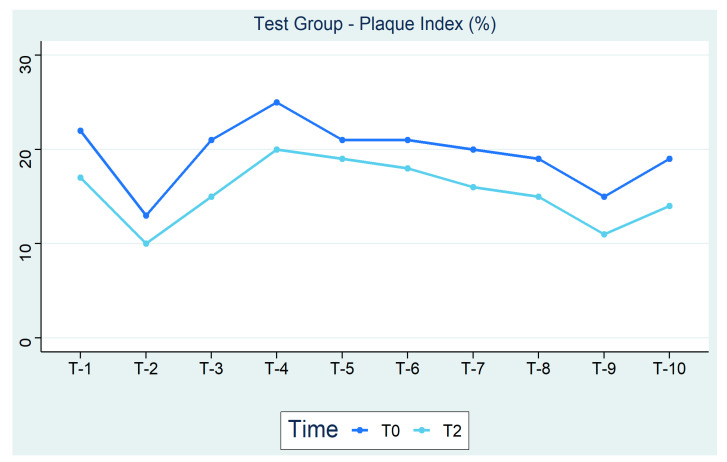
Test Group—Plaque index (%).

**Figure 2 ijerph-18-00044-f002:**
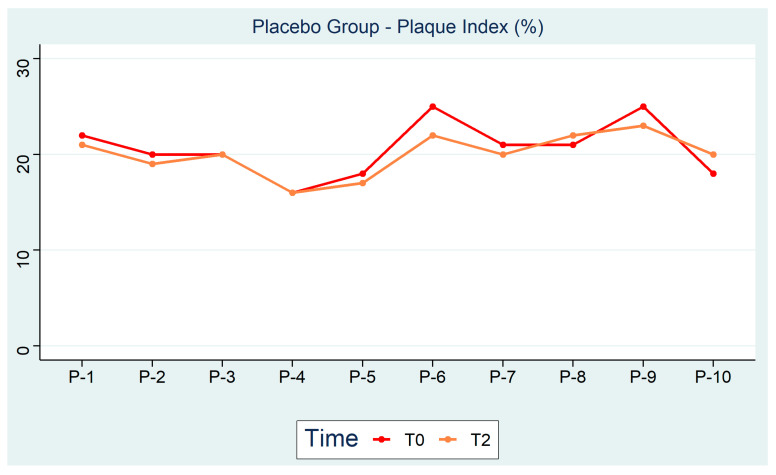
Placebo Group—Plaque index (%).

**Figure 3 ijerph-18-00044-f003:**
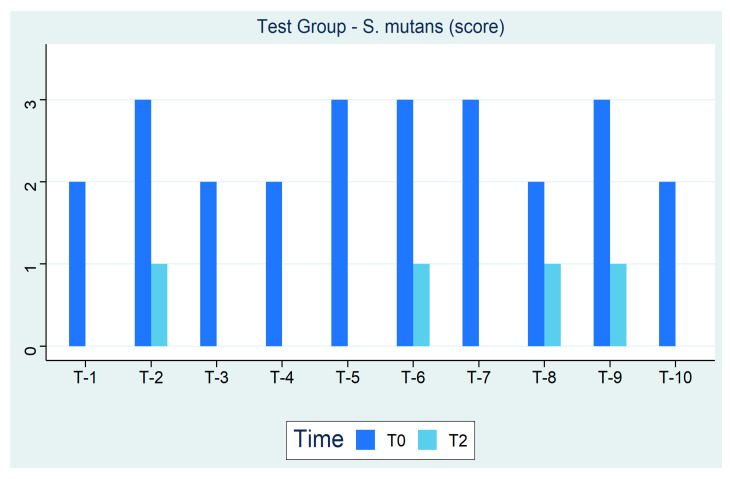
Test Group—*S. mutans* (Score).

**Figure 4 ijerph-18-00044-f004:**
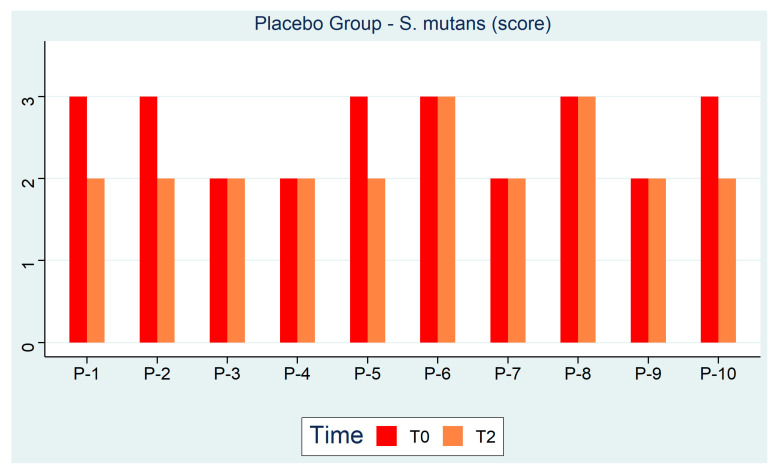
Placebo Group—*S. mutans* (Score).

**Table 1 ijerph-18-00044-t001:** Parameters for calculating *S. mutans* levels.

Score	CFU/ML	Caries Susceptibility
Class O	<10^3^	Negligible
Class 1	10^3^–10^4^	Low
Class 2	10^4^–10^5^	Moderate
Class 3	>10^5^	High

**Table 2 ijerph-18-00044-t002:** Plaque index in the Test group and the Placebo Group at different times.

Test Group Plaque Index (%)				Placebo Group Plaque Index (%)			
ID	T0 Baseline	T1 1 Month after the First Rinse	T2 3 Months after the First Rinse	ID	T0 Baseline	T1 1 Month after the First Rinse	T2 3 Months after the First Rinse
T-1	22	20	17	P-1	22	22	21
T-2	13	10	10	P-2	20	19	19
T-3	21	17	15	P-3	20	20	20
T-4	25	22	20	P-4	16	17	16
T-5	21	20	19	P-5	18	17	17
T-6	21	18	18	P-6	25	24	22
T-7	20	17	16	P-7	21	21	20
T-8	19	18	15	P-8	21	22	22
T-9	15	11	11	P-9	25	24	23
T-10	19	18	14	P-10	18	20	20

**Table 3 ijerph-18-00044-t003:** Different values of *S. mutans* score between the Study Group and the Placebo Group.

Test Group *S. mutans* (Score)				Placebo Group *S. mutans* (Score)			
ID	T0 Baseline	T1 1 Month after the First Rinse	T2 3 Months after the First Rinse	ID	T0 Baseline	T1 1 Month after the First Rinse	T2 3 Months after the First Rinse
T-1	2	2	0	P-1	3	3	2
T-2	3	2	1	P-2	3	3	2
T-3	2	1	0	P-3	2	2	2
T-4	2	1	0	P-4	2	2	2
T-5	3	1	0	P-5	3	3	2
T-6	3	2	1	P-6	3	3	3
T-7	3	2	0	P-7	2	2	2
T-8	2	2	1	P-8	3	3	3
T-9	3	2	1	P-9	2	2	2
T-10	2	2	0	P-10	3	3	2

**Table 4 ijerph-18-00044-t004:** Wilcoxon test *p*-values.

Test Group Plaque Index (%)	Placebo Group Plaque Index (%)	Test Group *S. mutans* (Score)	Placebo Group *S. mutans* (Score)
*p* < 0.001	*p* = 0.25	*p* < 0.01	*p* = 0.07

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
