# Peer review of "Efficacy of Sea Salt-Based Mouthwash and Xylitol in Improving Oral Hygiene among Adolescent Population: A Pilot Study"

_ijerph, 2020, doi:10.3390/ijerph18010044_

Round 1

Reviewer 1 Report

Suggestions have been made

Author Response

Thank you.

Reviewer 2 Report

The manuscript is much improved.   My comments are minor:

lines 162-163.   Usually one identifies adolescents as males/females rather then men/women.   I believe that it would help improve the merits of the article.

Lines 191-2-3.  In Figure 3, there is no identification of the green/blue bars as seen in Figures 1 and 2.   There addition should improve the manuscript.

Author Response

We thank the Reviewer for the considerable attention and the valuable comments that certainly helped us to improve the quality of the present paper. We have revised the manuscript according to the Reviewers’ comments. Thank you.

Reviewer 3 Report

Many methodological biases exist

Please see my remarks

Author Response

We thank the Reviewer for the considerable attention and the valuable comments that certainly helped us to improve the quality of the present paper. We have revised the manuscript according to the Reviewers’ comments (underlined in dark yellow in the main text). Statistical analysis was revised from a Biostatistician and p-values added to the text. We also removed unclear sentences. Thank you very much for your attention and courtesy.

Reviewer 4 Report

The submission addresses an important problem on the efficacy of sea salt-based mouthwash and xylitol in improving dental tissues among adolescent population. However, there are several limitations.

As I suggested in the first review, in my opinion, the introduction section is very lengthy, and sometimes it does not refer to the topic of the publication.

In the Material and methods section, the authors have clearly written about participants and methods. But I didn’t find the information about examiners. In my opinion, this kind of information is necessary for such a study. Who did the examination? Dentist? How many? Was the examiner (s) calibrated?

There is also necessary to give some information about the procedure of examination: When the patient was examined – at what time? Did the patient eat or drink something before, did the patients make any hygienic procedures just before the examination? Or how many time after?  This information is are essential for bacterial level in salivary samples.

I would suggest the inclusion of the following works to increase the bibliographic:

Salivary Biomarkers and Oral Microbial Load in Relation to the Dental Status of Adults with Cystic Fibrosis. doi.org/10.3390/microorganisms7120692

The relationship between oral hygiene level and gingivitis in children. doi: 10.17219/acem/70417

Author Response

We thank the Reviewer for the considerable attention and the valuable comments that certainly helped us to improve the quality of the present paper. We have revised the manuscript according to the Reviewers’ comments (underlined in light yellow in the main text). Thank you.

Round 2

Reviewer 3 Report

Insignificant corrections are needed...

Reviewer 4 Report

Thank you very much for the correction of the manuscript